# Synthesis and Biological Properties of Alanine-Grafted Hydroxyapatite Nanoparticles

**DOI:** 10.3390/life13010116

**Published:** 2022-12-31

**Authors:** Bruna Carolina Dorm, Mônica Rosas Costa Iemma, Benedito Domingos Neto, Rauany Cristina Lopes Francisco, Ivana Dinić, Nenad Ignjatović, Smilja Marković, Marina Vuković, Srečo Škapin, Eliane Trovatti, Lidija Mančić

**Affiliations:** 1Department of Health and Biological Sciences,Universidade de Araraquara-UNIARA, CEP, Araraquara 14801-340, Brazil; 2Institute of Technical Sciences of SASA, P.O. Box 377, 11000 Belgrade, Serbia; 3Innovative Centre, Faculty of Chemistry, University of Belgrade, P.O. Box 51 Belgrade, Serbia; 4Jožef Stefan Institute, P.O. Box 3000 Ljubljana, Slovenia

**Keywords:** hydroxyapatite, alanine, amino acid, surface modification, cell adhesion, cell viability

## Abstract

**Simple Summary:**

Hydroxyapatite is a biomaterial widely used in regenerative medicine. For this application, it is necessary that this material has good biocompatibility and adhesion to the proteins of the organism where the material will be implemented. The association of hydroxyapatite with amino acids, such as alanine, is an interesting alternative to increase biocompatibility and cell adhesion. In this work, materials were obtained from the grafting of the amino acid alanine onto hydroxyapatite by three different methods and two different precursors of amino acids, and promising results regarding grafting and biological tests were found in the samples obtained by in situ grafting and simple mixing with alanine.

**Abstract:**

Hydroxyapatite attracts great attention as hard tissues implant material for bones and teeth. Its application in reconstructive medicine depends on its biocompatibility, which is in a function of composition and surface properties. The insertion of a protein element in the composition of implants can improve the cell adhesion and the osseointegration. Having this in mind, the proposal of this work was to develop L-alanine-grafted hydroxyapatite nanoparticles and to study their biocompatibility. Two L-alanine sources and three grafting methods were used for hydroxyapatite surface functionalization. The efficiency of grafting was determined based on X-ray powder diffraction, Fourier-transform infrared spectroscopy, thermal analyses, and field-emission scanning electron microscopy. The results indicated the formation of hydroxyapatite with 8–25 wt% of organic content, depending on the grafting method. Protein adsorption, cell adhesion, and viability studies were carried out to evaluate biological properties of grafted materials. The viability of MG-63 human osteoblastic cells following 24 h incubation with the alanine-grafted hydroxyapatite samples is well preserved, being in all cases above the viability of cells incubated with hydroxyapatite. The alanine-grafted hydroxyapatite prepared in situ and by simple mixture showed higher protein adsorption and cell adhesion, respectively, indicating their potential toward use in regenerative medicine.

## 1. Introduction

Hydroxyapatite (HAp) is a calcium phosphate material with unit cell formula Ca_10_(PO_4_)_6_(OH)_2_. Due to its chemical composition and crystal structure similar to bone and dental phosphates, HAp is considered to be of great interest for synthesis in pure and composite forms. There are many techniques for hydroxyapatite synthesis, such as solid-state [1], chemical precipitation [2], hydrolysis [3], hydrothermal [4], pyrolysis [5], combustion [6], and electrospinning [7]. By choosing the method and the synthesis conditions, various morphologies were achieved, such as spheres, rods, needles, wires, etc. [8]. Besides pure hydroxyapatite, ions substitution was shown to be beneficial for achieving different HAp functionality. To increase similarity with natural bones, cations (such as Fe^3+^, Mn^2+^, Zn^2+^, Sr^2+^, Cu^2+^, Na^+^, etc.) and anions e.g., CO_3_^2−^, SiO_4_^4−^ and F^−^, were successfully incorporated in HAp without affecting its crystal structure, but influencing crystallinity, morphology, and solubility [9,10]. For better tracking of bone and dental implants behavior, doping with rare earth ions Yb^3+^, Tm^3+^, Eu^3+^, Gd^3+^, and Ho^3+^ was also applied [11,12,13]. It was also shown that HAp marked with radioactive isotopes like ^99m^Tc is a good indicator for osteogenesis process [14]. Besides favorable osteoinductivity [15] and osteoconductivity [16], there are a number of shortages associated with a low strength and toughness, brittleness, and poor mechanical resistance of HAp [17,18]. These lacks could be overcome by surface functionalization of HAp with different organic materials. For example, when poly-(lactic-co-glycolic acid) (PLGA) was used for HAp grafting, material with enhanced biocompatibility and increased tensile strengths was obtained [19]. Modification of HAp crystal surface toward increasing proteins uptake was accomplished by using bisphosphonates, polyphosphates, carboxyl groups, or pyrophosphate ions [20]. Lately, compounds that contain both amino and carboxylic acid functional groups were shown to be effective for achieving this goal [21,22]. For this purpose, composites made of HAp and PAG (polymers prepared from amino acids and glycolic acid) were proven to have excellent biocompatibility in vitro and in vivo [23]. Amino acids-grafted HAp solely, could be obtained by several manners, including the mixing of previously prepared HAp and amino acids [21], surfactant mediated approach [22], hydrothermal treating [24,25], precipitation with HAp seed crystals [26], and in situ grafting [27]. The lack of data related to biocompatibility of L-alanine amino acid-grafted HAp nanoparticles, as well as the lack of comparison of grafting methods efficiency, was the motivation for exploring different grafting procedures and to test biological properties of obtained grafted materials. Three different procedures of grafting were applied without utilizing demanding polymerization or polycondensation reactions, i.e., simple mixing, thermal induction, and in situ, using two different L-alanine precursors. Results of detailed structural, chemical, morphological, and biological analyses served as an instrument for the revealing grafting efficiency and biocompatibility of obtained materials. The aim of this study was also to choose the best method for the grafting of nano HAp with alanine toward creating materials with optimized biological properties.

## 2. Materials and Methods

### 2.1. HAp Synthesis

All chemicals used were of analytical grade and purchased from Sigma Aldrich. Pure hydroxyapatite was synthesized using aqueous solutions A and B in accordance with the procedure described in [28], where solution A was a mixture of Ca(NO_3_)_2_ × 4H_2_O and NH_4_OH and solution B comprised H_3_PO_4_ and NH_4_OH. Solution B was preheated to 50 °C and solution A was added dropwise under stirring. Then, the heater was turned off and the mixture was stirred during next 24 h at room temperature, after which the obtained gel-like product was rinsed with distilled water three times through centrifugation (5000 rpm, 2 min), collected and freeze-dried.

### 2.2. Grafting Procedures

(a)Simple mixing

Simple mixing was performed by homogenization of L-alanine or L-alanine methyl ester hydrochloride with HAp nanopowder in a 1:1 mass ratio, subsequent vortexing for 20 min, rinsing through centrifugation (5000 rpm/2 min), and drying at 60 °C for 1 h. Obtained nanopowders are marked as SMA, for the sample prepared with alanine precursor, and SME—for the sample prepared with ester precursor.

(b)Thermal induction 

In the thermal induction procedure, the same reactants as described above were homogenized in a 2:1 mass ratio, and after 20 min of vortexing, obtained mixtures were subjected to the heat treatments at 200 °C (1 h) in the case of L-alanine, and 140 °C (1 h) in the case of L-alanine methyl ester hydrochloride. After heating, nanopowders are naturally cooled down to room temperature and rinsed once through centrifugation (5000 rpm/2 min). Drying is performed at 60 °C for 1 h. Obtained samples are marked as TTA and TTE, depending on the used alanine precursor.

(c)In situ grafting

The in situ grafting procedure was performed according to the stated steps for pure hydroxyapatite synthesis, with the addition of L-alanine or L-alanine methyl ester hydrochlorideprecursor to the solution A in 1:1 mass ratio. Obtained samples are marked as ISA and ISE, depending on the used alanine precursor.

### 2.3. Characterization

#### 2.3.1. Powder Characterization

All synthesized powders were analyzed by X-ray powder diffraction (XRPD) in a range of 10–70° (Philips PW 1050); Fourier-transform infrared spectroscopy (FTIR) in the spectral range from 400 to 4000 cm^−1^ (Nicolet iS10 FTIR Spectrometer, Thermo Scientific, Madison, MI, USA); thermogravimetric analyses (TG/DTA, DSC), up to 500 °C, with a heating rate of 10 °C/min in air atmosphere (SETSYS Evolution 24,000 Setaram Instrumentation, Cailure, France); differential scanning calorimetry up to 250 °C, with heating rate of 10 °C/min in N_2_ atmosphere (Evo 131 Setaram Instrumentation, Cailure, France); and field-emission scanning electron microscopy (Carl Zeiss ULTRA Plus FESEM, Oberkochen, Germany). 

#### 2.3.2. Protein Adsorption

The protein adsorption was determined by incubating the samples (5 or 10 mg) in 1 mL of albumin solution (0.25 mg/mL, in distilled water, pH 7.4) for 10 and 30 min at room temperature. After the incubation, the samples were centrifuged at 5000 rpm for 5 min. The protein concentration in the supernatant was determined by UV absorption at 280 nm using a Bel UV-Vis (V-M5) spectrophotometer. The protein adsorbed at the surface of the samples was calculated from the difference between the initial and the final mass of protein in solution and the results were shown in percentages, using Excel software for plotting[29].

#### 2.3.3. Cell Adhesion

Samples (10 mg) were spread on the surface of a round coverslip (1.3 cm diameter), glued with superglue (3M) and sterilized under UV radiation for 30 min. MG-63 human osteoblastic cell line (purchased from Rio de Janeiro Cell Biobank, Brazil) was used to study the cell attachment to the samples and cell viability assay. The cells were maintained in Dubelcco’s Modified Eagle´s Medium—DMEM with 10% of fetal bovine serum (Nutricell, Gibco/Thermofisher, Waltham, USA) and antibiotics (penicillin 100 U/mL; streptomycin 0.1 mg/mL, from Sigma Aldrich, München, Germany) in a humidified atmosphere of 5% CO_2_ and 95% air at 37 °C. For the cell adhesion assay, the samples (coated coverslip) were inserted into the wells; the cells (5 × 10^5^ cells per well, in 1 mL of DMEM supplemented culture medium) were seeded on the samples and cultured in 5% CO_2_ and 95% air atmosphere at 37 °C. After culturing the cells for 24 h, each sample was rinsed with phosphate buffer saline (PBS), then fixed with paraformaldehyde (2% vol in PBS) at room temperature for 20 min and washed with PBS three times. Each sample was stained with 40,60-diamidino-2-phenylindole (DAPI, Merck, Billerica, MA, USA) for 10 min, for nuclei labeling. The cells, in five random fields on each sample, were observed using a fluorescence microscope (Leica MDi8, Wetzlar ,Germany). 

#### 2.3.4. Cell Viability

Cell viability was investigated using the standard resazurin reduction method. The cell culture was maintained at 37 ± 2 °C in 5% CO_2_ atmosphere and trypsinized when they reached confluence of 80–90%. The cell suspension was then centrifuged for 5 min at 1200× *g*, and 1 mL (1 × 10^6^ cells) was seeded into a 24-well plate filled with the coated coverslip (samples). The cells were also seeded into an empty well and cultivated at the same conditions and used as the negative control (to ensure the cell growth free of sample). At predetermined time points (24 and 48 h), the culture medium was removed and the cells were washed with PBS. Resazurin solution (10% vol in DMEM-supplemented medium) was added to each well and the microplates were incubated at 37 ± 2 °C for 4h and 5% CO_2_ atmosphere. An amount of 100 µL of each sample was transferred to the 96-well microplate. Resazurin was removed and the wells were filled with 1mL of culture medium and incubated again at the culture conditions. The fluorescence was read at 570 nm excitation and 590 nm emission wavelengths in a microplate reader (CaryEclipse Agilent Technologies, Santa Clara, CA, USA). The cell viability was reported as fluorescence intensity, using Excel software for plotting. The results are reported as mean values ± SD. 

## 3. Results

### 3.1. Structural, Morphological and Physic-Chemical Characterization

XRPD patterns of HAp before and after grafting are presented in Figure 1. As indicated by indexed Bragg reflections, identified maxima in all samples belongs to the hexagonal hydroxyapatite which crystallizes in *P6_3_/m* space group (JCPDS No. 89-6440). Evident broadening of the reflections is associated with nanocrystalline size of HAp particles, while the emergence of a weak reflection at 20.67° in TTA diffractogram implicates presence of L-alanine (JCPDS No. 28-1508). According to the Debye–Scherrer equation, the determined crystallite size of Hap, SMA, SME, and TTA is 30 nm. A slight decrease in crystallite size to 25 nm in TTE sample is provoked by heat treatment of powder at a temperature which is higher than the melting temperature of L-alanine methyl ester hydrochloride, so Cl^−^ ions from the melt adhere to the HAp surface, inhibiting further crystal growth. For ISA and ISE samples, the smallest crystallite size of 18 nm is calculated. According to the literature, dissolved amino acids could chelate Ca^2+^ and PO_4_^3−^ ions and bind to the Hap nuclei, inhibiting their further growth [24,30]. Although Tanaka et al. have shown that alanine establishes the weakest interaction with HAp in comparison to other amino acids [31], it is obvious that the level of interaction achieved here was sufficient to suppress the crystallites growth in ISA and ISE samples. 

HAp grafting with alanine was confirmed by FTIR analysis, as shown in Figure 2. The following vibrational modes of HAp were identified in spectra: PO_4_^3−^ ν4 out of plane bending at 550 and 598 cm^−1^, PO_4_^3−^ ν1 symmetric stretching at 1006 cm^−1^, OH^−^ stretching at 3570 cm^−1^, and adsorbed water stretching at 1634 and 3320 cm^−1^ [32]. In addition, L-alanine shows characteristic vibrational modes of NH_2_ bending at 1309 cm^−1^, CH_3_ bending at 1421 and 1447 cm^−1^, NH_3_ bending and stretching at 1585 and 2976 cm^−1^, respectively, CH_3_ stretching at 2600 cm^−1^ and H_2_O at 1600 cm^−1^ [33,34,35]. The characteristic L-alanine mode of NH_2_ bending at about 1300 cm^−1^ is present in spectra of all grafted samples. Its intensity is strongest for the SMA sample. A few other L-alanine modes are detectable in spectra of grafted samples but with weaker intensity. For all grafted samples, the presence of adsorbed H_2_O is prominent.

The FTIR spectrum of methyl ester hydrochloride, presented together with spectra of HAP, SME, TTE, and ISE in Figure 3, shows that besides characteristic modes of L-alanine (NH_3_ and CH_3_), this compound has additional modes of CH bending at 1229 cm^−1^ and C=O bending at 1740 cm^−1^ [33,36]. As it can be seen, all of these are present in grafted samples-related spectra too, but with weaker intensities. 

Comparative thermal analyses of the samples were also performed in order to evaluate the level of grafting, shown in Figure 4 and Figure 5. As observed in Figure 4a, thermal decomposition of L-alanine occurs at about 310 °C, which corresponds well to the literature [37,38,39]. A similar temperature range is observed for L-alanine methyl hydrochloride decomposition (Figure 4b). Both alanine precursors exhibit higher weight loss than HAp in the same temperature range, so the differences among detected weight loss in HAp (of about 5 wt%) and grafted samples were attributed to the quantity of alanine retained in these. Considering simple mixing, both samples decrease in their weight for about 8 wt%. Weight loss of 16 wt% observed for the TTA sample corresponds well to the twice larger mass of alanine precursor used for grafting through thermal induction. On the other hand, a higher content of alanine is present in TTE, judging by the 25 wt% loss observed. The same trend is detected for ISA and ISE, where losses of 10 and 14 wt% were measured.

The results of the DSC analyses, shown in Figure 5, implicate that desorption of water near 100 °C is the most prominent endothermic process caused by grafting. Judging from the steepness of the slopes, it is obvious that water release is slower in grafted samples obtained by thermal induction than in their counterparts obtained through simple mixing. Moreover, the process of water desorption in the ISA and ISE samples takes place at higher temperatures than in others, meaning that more energy is required for this process. 

Modification of the HAp powders morphology due to grafting is revealed by FESEM analysis, shown in Figure 6 and Figure 7. The well-dispersed rod-like HAp particles with an average length of about 100 nm and diameter of 10 nm are clearly distinguished in Figure 6. The Ca/P ratio of 1.5 calculated directly from the EDS result is lower than the theoretical value of 1.67, so Ca-deficient HAp is obtained during precipitation. With grafting, HAp particles become more prone to agglomeration, but without change in their morphology. 

It is evident that agglomeration increases gradually in the following way: simple mixing, thermal induction, and in situ grafting, and is generally more intense in all samples obtained through grafting with L-alanine methyl ester hydrochloride. This indicates a higher level of reactivity of L-alanine methyl ester hydrochloride for grafting, in comparison to L-alanine. 

### 3.2. Protein Adsorption

Figure 8 shows the results of the protein adsorption test. The interactions between cells and grafted HAp particles were measured after 10 and 30 min. When 5 mg of grafted samples were tested after 10 min, the beginning of the protein adsorption at the particles’ surfaces is observed. With the prolongation of time to 30 min, the mass of protein adsorbed to the surface of the SME, ISA, and ISE samples abruptly increased, reaching the maximum of ~35% for ISE. Doubling the samples’ mass (10 mg) also enhances adsorption in these samples for both tested times. The mass of protein adsorbed after 30 min at SME, ISA, and ISE surface reached 33, 36, and 50 (%wt), respectively. The negative values observed for HAp, SMA, TTA, and TTE with prolongation of time indicate that these nanoparticles remain trapped in protein solutions due to the formation of stable suspension which could not be removed by centrifugation. The value obtained for the ISA at a concentration of 10 mg and time of 10 min was zero, since the measured absorbance value for this sample was the same as obtained for the standard albumin solution.

### 3.3. Cell Adhesion

The cell adhesion at the surface of the samples was visualized by DAPI labeling of their nuclei, as shown in Figure 9. The light-blue spots indicating the cells’ nuclei were found in HAp, SMA, SME, ISA, and ISE. The heat-treated samples TTA and TTE did not display any cell at their surface, confirming the results presented in Table 1. 

### 3.4. Cell Viability

The cell viability results determined by the resazurin test are shown in Figure 10. The metabolization of resazurin by live cells generates the fluorescent resorufin which is used to estimate the cell viability through measurement of the fluorescence intensity. Although lower than in the control, cell viability is preserved after 24 h hours in all samples. Furthermore, a clear tendency of cell growth after 48 h of incubation is confirmed for all samples, except for TTE which has been dispersed into the medium and was not read at the 48 h time point. 

## 4. Discussion

Alanine is the simplest molecule used as the constructive block of proteins, with one carboxylic acid, one amine group, and few carbons and hydrogens. Its grafting to HAp nanoparticles should contribute to the biocompatibility and osseointegration of implants based on them. However, to the best of our knowledge, the study of methods for grafting alanine to HAp, as well the study of the alanine-based HAP grafts’ cytotoxicity and cell adhesion was not described in the literature. Ca-deficient HAp rod-like nanoparticles with an aspect ratio of 10, obtained in this work by simple precipitation, have better crystallinity (crystallite size of 30 nm) than nanoparticles obtained through in-situ grafting (crystallite size of 18 nm). This implies that the presence of alanine (either in the form of acid or ester) during HAp precipitation suppresses its crystal growth. Inhibition of HAp growth in the presence of some other poly(amino acids) containing amine and carboxylate groups is already reported in literature, and is usually associated with the chelation of Ca^2+^ and PO_4_^3−^ ions by dissolved amino acids [40]. With grafting, HAp nanoparticles become more prone to agglomeration, but their morphology stays unchanged. The most intense agglomeration is characteristic of the ISA and ISE samples in which HAp nanoparticles were cross-linked in situ by alanine-forming porous nanostructures with low degrees of long-range order. For these, 10 and 14 wt% of alanine is calculated based on TGA. A somewhat smaller alanine content of 8 wt% is detected in samples obtained by simple mixing, while 25 wt% of alanine is estimated from the weight loss of the TTE sample. The presence of the characteristic modes of L-alanine (NH_3_ and CH_3_) is confirmed in all grafted samples by FTIR spectroscopy. The characteristic L-alanine mode of NH_2_ bending at about 1300 cm^−1^ is strongest for the SMA sample, while additional modes of CH bending at 1229 cm^−1^ and C=O bending at 1740 cm^−1^ are most pronounced in spectra of the SME and ISE samples. Their presence on the HAp nanoparticles surface provides additional binding sites, which is confirmed by the protein adsorption test. Namely, an increase of about 50wt% for protein adsorption at the SME and ISE surfaces is detected with the prolongation of time from 10 to 30 min, shown in Table 1. The capability of binding body proteins from blood serum is an extremely important property of biomaterials in general [41]. When the artificial biomaterial is inserted into the body, the first response of the body is to trigger the innate protective immune system, i.e., to cover its surface with proteins from blood serum. Consequently, the protein adsorption rapidly changes the biomaterial surfaces influencing inflammatory response and the wound-healing process in the body. The results of protein adsorption on SME, ISA, and SME (Figure 9) indicate that their surface was fast-coated by the proteins. The results of the cell adhesion test (Figure 10) corroborate with the protein adsorption test, evidenced by microscopic images showing the cells adhered at the surface of Hap, SMA, SME, ISA, and ISE. In general, these results indicate the potential of SMA, SME, ISA, and ISE for use as grafts, and these samples could be considered as the most promising for application in bone grafts, for humans and animals, once the cell adhesion is crucial for osseointegration. Beside adhesion, the cytotoxicity test is the most frequently utilized for ensuring the safety of biomaterial use. As shown in Figure 10, the viability of MG-63 human osteoblastic cells following 24 h incubation with the alanine-grafted HAp samples is well preserved, being in all cases above the viability of cells incubated with HAp. The viability of cells was even higher after 48 h of exposure in all cases, excluding ISE. The most intense cell proliferation with a prolongation of the exposure is observed for cells incubated with SMA. A similar trend of MG-63 cell growth is determined after 3 days of their incubation with composites based on HAp and aspartic/glutamic acid [42]. The same study shows that composites were able to better support the growth of MG63 cells during the long-time exposure of 7 days, contributing to osteoblast activation and extra-cellular matrix mineralization processes. 

Keeping in mind that the appropriate cellular response to biomaterials is essential for tissue regeneration [43], good viability of the MG-63 human osteoblastic cell and superior protein adsorption at the surface of samples obtained through in situ grafting reflect promising cell adhesion, which is a prerequisite for successful osseointegration.

## 5. Conclusions

Presented results demonstrate successful grafting of alanine to HAp nanoparticles by all synthesis methods explored. Two alanine precursors were used for grafting: pure L-alanine and L-alanine methyl ester hydrochloride. The higher content and uniform distribution of alanine in samples obtained through in situ grafting contribute to higher protein adsorption and good MG-63 human osteoblastic cell viability. Based on this, these samples could be considered as the most promising for further investigation toward their potential use in regenerative medicine.

## Figures and Tables

**Figure 1 life-13-00116-f001:**
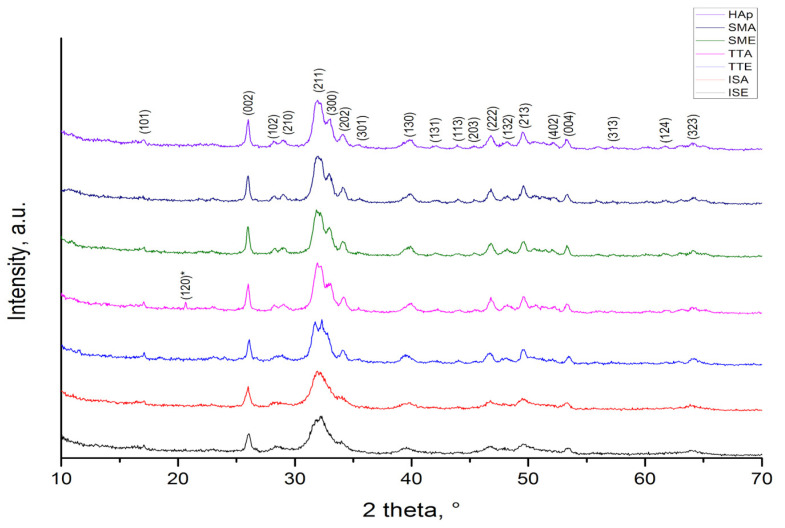
XRPD patterns of HAp and HAp grafted samples.

**Figure 2 life-13-00116-f002:**
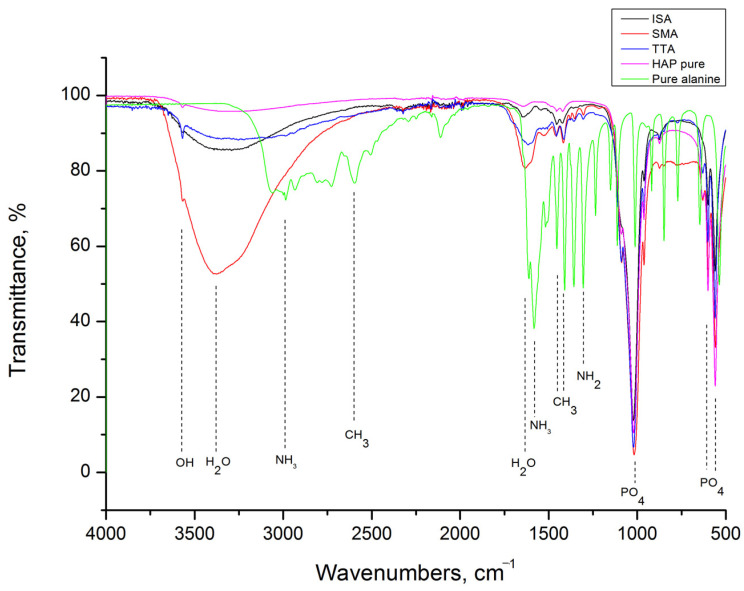
FTIR spectra of HAP, L-alanine, SMA, TTA, and ISA samples.

**Figure 3 life-13-00116-f003:**
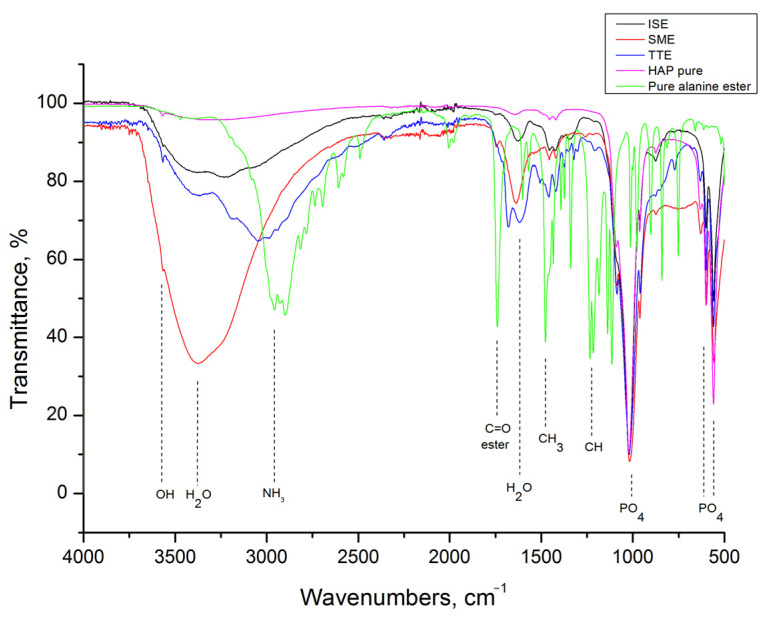
FTIR spectra of HAP, L-alanine methyl ester hydrochloride, SME, TTE, and ISE samples.

**Figure 4 life-13-00116-f004:**
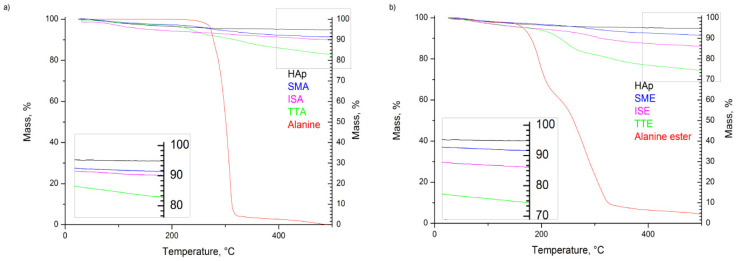
TGA diagrams of: (**a**) SMA, TTA, ISE; (**b**) SME, TTE, ISE grafted samples.

**Figure 5 life-13-00116-f005:**
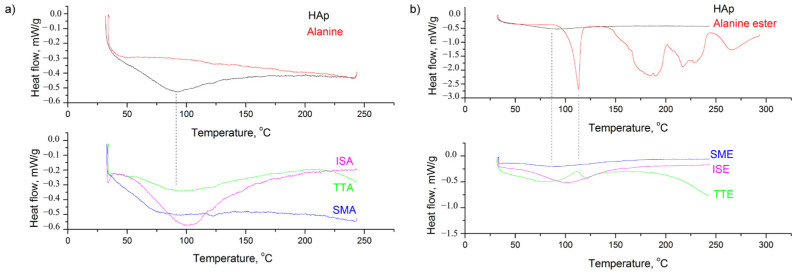
DSC diagrams of samples grafted with: (**a**) L-alanine; (**b**) L-alanine methyl ester hydrochloride.

**Figure 6 life-13-00116-f006:**
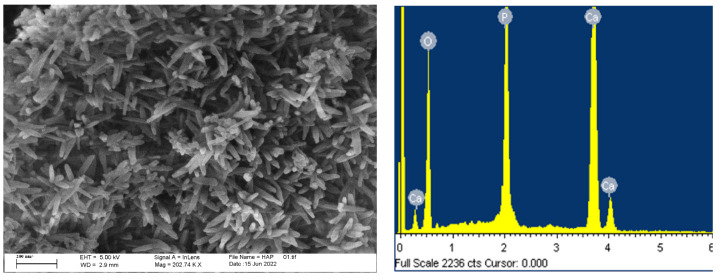
FESEM micrograph of HAp powder and corresponding EDS analysis.

**Figure 7 life-13-00116-f007:**
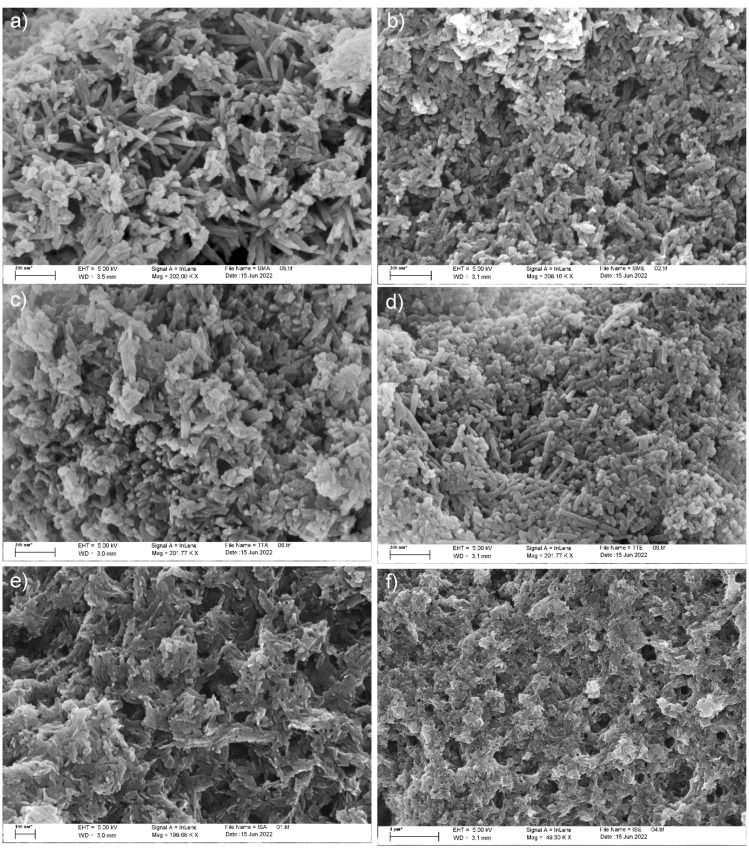
FESEM micrographs of grafted samples: (**a**) SMA; (**b**) SME; (**c**) TTA; (**d**) TTE; (**e**) ISA; (**f**) ISE.

**Figure 8 life-13-00116-f008:**
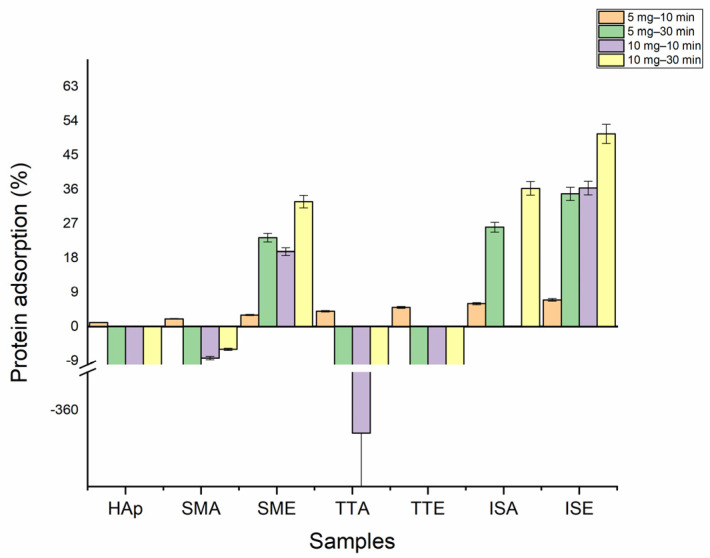
Protein adsorption test.

**Figure 9 life-13-00116-f009:**
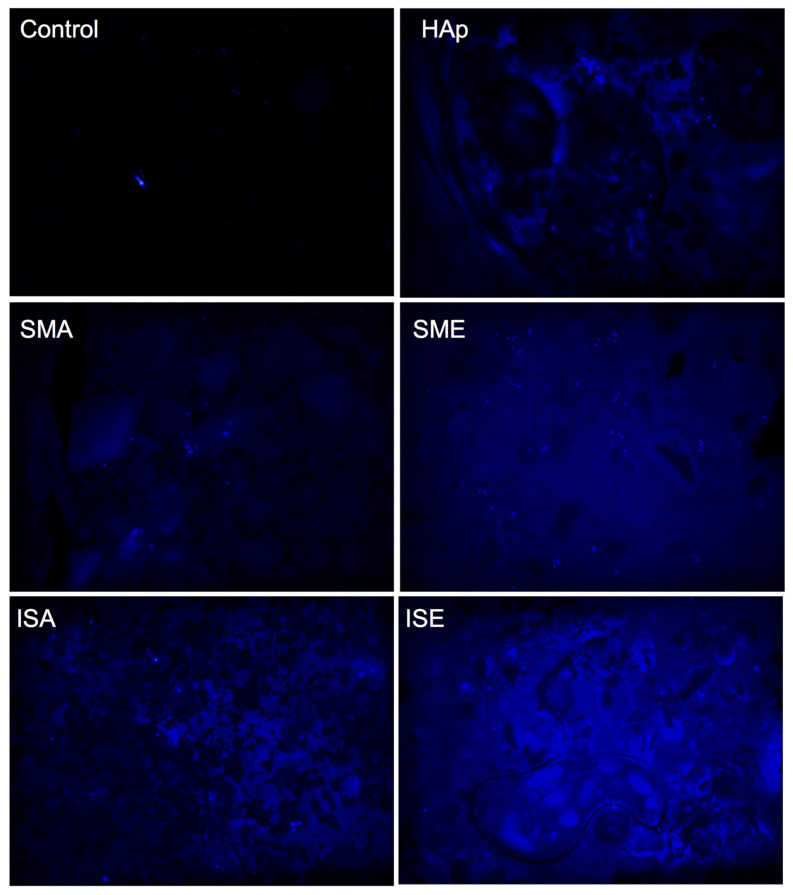
Cell adhesion at the surface of pure HAp and alanine-grafted HAp samples. The visualization of cells’ nuclei with DAPI staining is shown as light-blue spots at samples’ surface. The blurred blue background is due to autofluorescence of samples. Control shows cells at the surface of well without sample.

**Figure 10 life-13-00116-f010:**
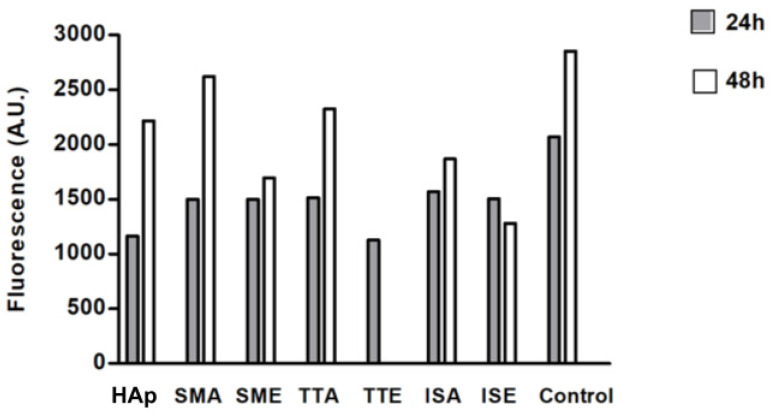
Cells viability at the surface of the samples.

**Table 1 life-13-00116-t001:** Protein adsorption (%) values.

Protein Adsorption (%)				
Sample	5 mg–10 min	5 mg–30 min	10 mg–10 min	10 mg–30 min
HAp	−20.16	−50.08	−147.4	−118.36
SMA	−8.12	−16.75	−8.28	−5.96
SME	14.15	23.32	19.68	32.8
TTA	−279.92	−280.16	−366	−329.8
TTE	−48.28	−25.72	−98.8	−84.48
ISA	16.2	26.08	0	36.32
ISE	26.16	34.88	36.4	50.6

## Data Availability

Not applicable.

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
