# Peer review of "Synthesis and Biological Properties of Alanine-Grafted Hydroxyapatite Nanoparticles"

_life, 2022, doi:10.3390/life13010116_

Round 1

Reviewer 1 Report

Dear authors,

the present manuscript is an interesting research but somehow the novelty is not enough presented. Please state clear in abstract and introduction about the need and what brings out teh study. A short paragraph about future perspectives shall be included in discussions section to have a more logical flow and related to the sentence from conclusions section: " these samples could be considered as the most promising for application in
bone graft".

Lines 67-70: what do you mean by " modesty of data related to success"?! there are no references to support this statement.

In the discussions section there is no link to existing research or " modesty of data", plaese re-consider this section with relevant supported references to show a clear progress compared to the state-of-art.

Reviewer 2 Report

Dorm et al. performed and interesting study and showed that alanine grafted HAp prepared in situ and by simple mixture participate in cell adhesion and can be used as a graft.

The article is well written and organised. The Materials and methods Section provides sufficient data and the results are clearly presented. I have only minor suggestions.

The authors should mention what software used for figures, such as Figure 8 or Figure 10.

In the Discussion section the authors should add more data on the medical implications of the results of their study.
